# Development of the Direct Deuteration Method for Amino Acids and Characterization of Deuterated Tryptophan

**DOI:** 10.3390/bioengineering12090981

**Published:** 2025-09-16

**Authors:** Chie Shibazaki, Haruki Sugiyama, Misaki Ueda, Takayuki Oku, Motoyasu Adachi, Zoë Fisher, Kazuhiro Akutsu-Suyama

**Affiliations:** 1J-PARC Center, Japan Atomic Energy Agency (JAEA), 2-4 Shirakata, Tokai, Naka, Ibaraki 319-1195, Japan; takayuki.oku@j-parc.jp; 2Neutron Industrial Application Promotion Center, Comprehensive Research Organization for Science and Society (CROSS), 162-1 Shirakata, Tokai, Naka, Ibaraki 319-1106, Japan; h_sugiyama@cross.or.jp; 3Neutron Science and Technology Center, Comprehensive Research Organization for Science and Society (CROSS), 162-1 Shirakata, Tokai, Naka, Ibaraki 319-1106, Japan; m_ueda@cross.or.jp; 4Institute for Quantum Life Science, National Institutes for Quantum Science and Technology (QST), 4-9-1 Anagawa, Inage, Chiba 263-8555, Japan; adachi.motoyasu@qst.go.jp; 5European Spallation Source ERIC, 221 00 Lund, Sweden; zoe.fisher@ess.eu; 6Lund Protein Production Platform, Department of Biology, Lund University, Sölvegatan 35, 22362 Lund, Sweden

**Keywords:** amino acid, deuteration, optical isomer, crystal, isotope effect

## Abstract

Proteins and peptides are vital biomolecules, and deuterated amino acids are increasingly applied in areas such as drug discovery, metabolic tracing, and neutron scattering studies. In this study, we performed deuteration on all 20 proteinogenic amino acids, including their side chains, and established efficient methods for 13 amino acids. Using a Pt/C-catalyzed hydrogen–deuterium exchange reaction, the reaction parameters were optimized to achieve the selective and stable incorporation of deuterium. In addition, the resulting deuterated compounds, focusing on tryptophan, were characterized in order to assess their physicochemical properties. Because the deuteration reaction caused significant racemization of amino acids, deuterated D/L-tryptophan was isolated using a chiral separation method. Deuterated tryptophan characterization studies confirmed that the photostability was markedly enhanced by deuteration, whereas the acid stability showed no clear isotopic effect. The X-ray crystal structure analyses revealed minimal changes upon the hydrogen-to-deuterium substitution. These results provide a robust platform for the supply of deuterated amino acids, facilitating their application in drug development, structural analysis, and creation of advanced functional biomaterials.

## 1. Introduction

Proteins and polypeptides are fundamental components of biological structures such as muscles, skin, and internal organs, and they play essential roles in maintaining cellular functions. These macromolecules function in diverse physiological processes such as enzymes, hormones, antibodies, and other bioactive agents, contributing to digestion, metabolism, immunity, and intercellular signaling. Malfunctions or structural abnormalities in biomolecules, particularly those involved in physiological activity, are known to cause various diseases, making them critical targets for biomedical research and drug discovery [1,2,3].

Amino acids, the building blocks of proteins, form polypeptides through specific sequences of 20 standard types, each contributing to the final three-dimensional structure and function of the protein. In addition to their structural roles, amino acids serve as precursors of neurotransmitters and hormones. For example, glutamic acid functions as an excitatory neurotransmitter and is a known precursor of γ-aminobutyric acid (GABA) and a key intermediate in amino acid metabolism [4]. Similarly, tryptophan is a biosynthetic precursor of serotonin and melatonin compounds involved in the regulation of mood, sleep, and circadian rhythms, making it an important modulator of physiological functions in the central nervous system [5,6].

Metabolic tracing and pharmacokinetic analysis are essential to elucidate how nutrients and bioactive compounds are metabolized in vivo and to clarify how metabolic pathways and intermediates are altered in conditions such as cancer or metabolic disorders. In this context, deuterium-labeled amino acids, in which hydrogen atoms are replaced with deuterium atoms, have proven to be effective tracers [7]. Moreover, the pharmaceutical field has seen growing interest in the use of deuterium isotopes to enhance drug performance [8,9]. Because carbon–deuterium (C–D) bonds are stronger than carbon–hydrogen (C–H) bonds, they confer greater metabolic and photochemical stability, which can improve the pharmacokinetics of drugs that are rapidly degraded in vivo. This has led to increasing interest in the development of deuterated drugs.

The structural and functional analyses of biomolecules often rely on X-ray scattering and diffraction techniques. Neutron scattering, a well-established technique, has in recent years emerged as a powerful complementary method in structural biology, biomolecular dynamics, and deuterated sample studies, owing to advances in accelerators, neutron sources, and sample preparation techniques [10,11]. By substituting hydrogen atoms in a sample with deuterium, researchers can exploit the differences in neutron scattering cross sections to reduce background noise and enhance contrast in specific regions [12]. Furthermore, site-selective deuteration enables the acquisition of high-resolution information on conformational dynamics and structural changes in targeted domains [13].

Currently, several commercial routes are available for the production of deuterated amino acids. One widely used approach is chemical synthesis, in which deuterated reagents or solvents such as D_2_O, CD_3_OD, or DCl are incorporated during the synthesis of amino acids or their precursors. This method typically requires multiple steps and can be associated with high production costs [14,15]. Fermentation-based methods have also been developed, in which microorganisms are cultivated in deuterated media containing D_2_O or deuterated carbon sources (e.g., D-glucose). This approach allows the production of highly deuterated amino acids and is commonly used for generating isotope-labeled biomolecules for NMR spectroscopy or neutron scattering studies; however, it presents challenges in terms of cultivation cost and optimization of growth conditions [16,17]. Enzyme-catalyzed reactions in deuterated solvents offer another strategy, providing high positional selectivity for site-specific deuterium incorporation. These methods are often employed as complementary approaches for partial deuteration [18,19,20].

In pursuit of these applications, recent studies have demonstrated that ruthenium catalysts are effective for deuterating both the main chain and side chains of specific amino acids [21,22]. Ruthenium catalysts appear to successfully catalyze the deuteration of alpha hydrogen atoms in amino acids while preventing racemization. Organocatalysts accelerated the effective deuteration of alpha hydrogen atoms in amino acids at room temperature through racemization [23]. In the case of enzymic reaction, alpha and beta hydrogen atoms in amino acids were rapidly deuterinated by an enzyme without racemization [18,19,20]. In addition, Sajiki et al. developed H–D exchange reactions that introduce deuterium into amine and carboxylic acid compounds by replacing the C–H bonds with C–D bonds using palladium or platinum on carbon catalysts [24,25]. In light of these findings, it can be suggested that palladium or platinum on carbon-catalyzed deuteration reactions are effective for the deuteration of main chain and side chains of amino acids. Note that palladium or platinum in carbon-catalyzed deuteration reactions racemizes optical isomers [26] and may racemize amino acids as well (Table 1).

Despite these aforementioned advancements, the synthesis of deuterated compounds, especially those involving side-chain deuteration, remains expensive and technically challenging. Moreover, reports on the enantioseparation (chiral resolution) and physicochemical characterization of deuterated amino acids are limited.

In this study, we aimed to develop a practical and efficient method for the selective deuteration of 20 proteinogenic amino acids, including their side chains, and to establish a protocol for enantiomeric separation. We evaluated the physicochemical properties of the resulting compounds, including their stability under ultraviolet (UV) irradiation and acidic conditions, to explore their potential use as high-performance biomaterials. Through this study, we aim to contribute to the creation of a sustainable and cost-effective supply platform for deuterated amino acids with applications in drug development, structural biology, and materials science.

Particular attention has been paid to tryptophan, an amino acid of significant interest because of its role as a serotonin and melatonin precursor. Given its relevance in the fields of nutraceuticals and pharmaceuticals, our study focused on the enantioseparation crystal structure analysis, and physicochemical characterization of deuterated tryptophan.

## 2. Materials and Methods

### 2.1. Samples and Reagents

Amino acids (TCI Chemicals Co., Ltd., Tokyo, Japan), activated carbon (Osaka Gas Chemical Co., Ltd., Osaka, Japan, Figure 1), Platinum on Carbon catalyst (Pt/C) (3 wt% Pt, Type STD (wetted with water), N.E. CHEMCAT Co., Tokyo, Japan), 2-propanol (FUJIFILM Wako Pure Chemical Co., Tokyo, Japan), deuterium oxide (D_2_O) for deuteration and Nuclear Magnetic Resonance (NMR) solvent (99.9% D, Sigma-Aldrich, St. Louis, MO, USA), dimethyl sulfoxide (FUJIFILM Wako Pure Chemical Co., Tokyo, Japan), and methanol (FUJIFILM Wako Pure Chemical Co., Tokyo, Japan) were used without further purification. Ultrapure water (18.2 MΩ.cm) was produced with a deionized purified water production system (RFU424TA system, ADVANTEC, Tokyo, Japan) and used throughout this study.

### 2.2. Configuration of a Simplified Deuteration Apparatus Used for Deuteration

Deuterated amino acid synthesis was performed by sealing the reaction mixture in a small autoclave-type high-pressure reactor (100 mL capacity; Huanyu, Zhengzhou, China) equipped with an inner Teflon liner. The reactor was placed in an aluminum bead bath (inner diameter: 130 mm, depth: 160 mm; Tokyo Rikakikai Co., Ltd., Tokyo, Japan) filled with aluminum beads (Tokyo Rikakikai Co., Ltd., Tokyo, Japan) and heated using a hot-plate magnetic stirrer (RCH-1000, Tokyo Rikakikai Co., Ltd., Tokyo, Japan).

### 2.3. Procedure for Deuteration of Amino Acids

A mixture of amino acids (1 g) and Pt/C (3 wt% Pt, 0.40 g, 0.06 mmol) in 2-propanol (4 mL)/D_2_O (40 mL) was loaded into the reactor. The mixture was then heated to 100–230 °C and stirred continuously for one to several days. After cooling to 20 °C, the Pt/C catalyst was then removed by celite filtration and further filtered through a 0.22 μm filter. The filtrate was evaporated to dryness under reduced pressure to obtain deuterated amino acids. For some amino acids, impurities produced by deuteration reactions were removed by washing the crude product with ethanol.

A blank test was conducted with glycine, and the obtained glycine was 5.5% deuterated (See Appendix A). This result indicates that Pt/C (and Pd/C) exhibits a key catalytic effect on the deuteration of glycine and other amino acids.

### 2.4. Analytical Methods Using NMR to Determine the Deuteration Level of Amino Acids

The ^1^H and ^2^H NMR spectra of the deuterated amino acids were recorded using a 400 MHz NMR spectrometer (JEOL JMTC-400/54/JJ/YH spectrometer, ^1^H: 400 MHz, ^2^H: 61.4 MHz) to confirm the deuteration level of the amino acids. Details of the sample preparation method for NMR measurements are provided in the Appendix A. Note that the results of quantitative NMR studies suggested that it is necessary to pay close attention to the integration range of NMR peaks, the weighing of samples, and so on in order to analyze with an error of less than 1% [27]. Since we performed NMR analysis without paying special attention, it is reasonable to assume that the error in this study is not less than 1%.

### 2.5. Measurement of Optical Rotation

The specific rotation of the samples was measured using a polarimeter (Model P-2200, JASCO, Tokyo, Japan) modified for acid resistance and equipped with a Peltier thermostated cell holder (PTC-262). The measurements were performed at 20 °C using a cylindrical quartz cell (100 mm × 3.5 mm, optical path length: 10 cm) at a wavelength of 598 nm. Each measurement was performed five times and the average value was used. The observed optical rotations were measured under identical conditions, and the specific rotation [α] D^20^ values were calculated accordingly. All of the measurements were conducted in accordance with standard polarimetric procedures. Commercial L-tryptophan (L-Trp) was dissolved in distilled water at a concentration of 0.9690% (*w*/*v*). The solution was heated to 80 °C to ensure complete dissolution before measurement. Deuterated L-tryptophan was dissolved in distilled water at a concentration of 0.56277% (*w*/*v*). Complete dissolution was achieved by heating the solution to 90 °C.

### 2.6. Liquid Chromatography Setup for Enantiomeric Separation

High-performance liquid chromatography (HPLC) analysis was conducted using a HITACHI Chromaster system equipped with a Hitachi 5440AD fluorescence detector (excitation at 280 nm and emission at 350 nm, HITACHI, Tokyo, Japan). Enantiomeric separation was performed using a DAICEL CHIRALPAK ZWIX(+) column (4.0 mm I.D. × 250 mm L, 3 µm particle size) at a flow rate of 0.15 mL/min. The mobile phase consisted of methanol, acetonitrile, and water at a ratio of 39:39:22 (*v*/*v*/*v*), supplemented with 40 mM formic acid and 20 mM diethylamine. Commercial L-tryptophan (L-Trp) was dissolved in water by heating at 80 °C to prepare a 20.0 mM stock solution. This solution was diluted 100-fold with the eluent, yielding a final concentration of 0.2 mM. An injection volume of 1 µL, corresponding to 2.0 pmol of L-Trp, was used for the HPLC analysis. Deuterated tryptophan (d-Trp) was prepared by dissolving water at 90 °C to obtain a 20.0 mM stock solution. Following 100-fold dilution with the eluent, the final concentration was 0.2 mM. A 1 µL aliquot, equivalent to 2.0 pmol of d-Trp, was injected into the HPLC system.

### 2.7. Fluorescence Measurement Method for Evaluating UV-Induced Degradation and Acid-Induced Changes

To measure the UV-induced degradation of tryptophan, samples were irradiated with UV light at 254 nm using a fluorescence spectrophotometer (FP-8300, JASCO Corporation, Tokyo, Japan). The intensity of the UV light was 62.94 mJ/min/cm^2^ at the center of the sample. Tryptophan fluorescence measurements were performed using an FP-8300 fluorescence spectrophotometer (Jasco, Heckmondike, UK) with an excitation wavelength of 290 nm and an emission range of 295–450 nm. For the analysis of the fluorescence intensity changes under acidic conditions, 6 M hydrochloric acid (FUJIFILM Wako Pure Chemical Co., Tokyo, Japan) was used, and the sample was heated at 80 °C while monitoring the fluorescence intensity at 339 nm.

### 2.8. Crystallization

Deuterated tryptophan (d-Trp), protonated tryptophan (h-Trp), deuterated tryptophan hydrochloride (d-TrpCl), and protonated tryptophan hydrochloride (h-TrpCl) crystals were prepared. A suitable crystal was mounted on a glass capillary and transferred to the rugged two-axis goniometer of a RIGAKU diffractometer with equipped with mirror monochromated Mo-Kα radiation (λ = 0.71073 Å) and HyPix-6000HEIC detector (Rigaku, Tokyo, Japan). The cell parameters were determined and refined, and the raw frame data were integrated using CrysAlisPro 1.171.43.98a (Rigaku OD, 2023). The details of the sample preparation method for X-ray crystallography measurements are described in the Appendix A.

## 3. Results and Discussion

### 3.1. Deuteration of 20 Proteinogenic Amino Acids

In this study, we developed an efficient deuteration method for 20 proteinogenic amino acids, including side chains, based on the procedure reported by Sawama et al. [25,28,29,30]. Commercially available L-amino acids (alanine, arginine, asparagine, aspartic acid, cysteine, glutamine, glutamic acid, glycine, histidine, isoleucine, leucine, lysine, methionine, phenylalanine, proline, serine, threonine, tryptophan, tyrosine, and valine) were used as starting materials. Each amino acid, along with Pt/C catalyst, 2-propanol, and D_2_O, was placed in a Teflon-lined high-pressure vessel for the reaction. The vessel was stirred to ensure temperature uniformity and then heated to conduct the reaction. The degree of deuterium incorporation was evaluated using nuclear magnetic resonance (NMR) spectroscopy. Note that to examine whether metal ions were present in the deuterated amino acids, X-ray fluorescence (XRF) analysis was conducted (the details of the XRF data are described in the Appendix A). The XRF spectra of deuterated glycine and tryptophan show no metal ion peak. Since the XRF analysis offers great sensitivity at the ppm level for simultaneous concentrations of several elements in targeted materials, it can be suggested that no contaminants were found in the deuterated amino acids.

Initially, deuteration reactions were carried out under standard conditions at 200 °C for 24 h for all 20 amino acids (Appendix A). Glycine (1), phenylalanine (6), and histidine (17) showed average deuteration levels exceeding 80%, with incorporation rates of 91.0%, 80.7%, and 82.5%, respectively. The α-position (backbone C–H) was almost fully deuterated, with deuteration rates of 91.0%, 96.8%, and 98.6%, respectively, suggesting near-complete deuteration of both backbone and side chain under these conditions. Proline (18) also achieved an overall average deuterium incorporation rate of 61.8%, suggesting that increasing the reaction temperature may further enhance the extent of deuteration.

In contrast, amino acids with aliphatic hydrocarbon side chains (alanine (2), valine (3), leucine (4), and isoleucine (5)) and the cyclic hydrocarbon proline (18) exhibited high backbone deuteration (97.0%, 96.5%, 96.5%, 97.5%, and 98.6%, respectively), but their side chains showed poor deuterium incorporation. These results suggest that efficient interaction between the α-carbon and the catalyst, whereas interaction with the side-chain carbon atoms appears insufficient. The remaining amino acids decomposed under the same conditions, as confirmed by NMR analysis.

To improve the overall deuteration efficiency, the reaction conditions were optimized by varying the temperature, employing additives, and testing different metal catalysts (Figure 2).

Alanine showed poor side-chain deuteration at 200 °C (2, Appendix A). Acid/base additives are known to enhance the deuteration reactions [31,32]. Therefore, the addition of acetic acid, sodium hydroxide, and ammonia (equimolar) to the alanine substrate was tested. Among these, ammonia most effectively improved side chain deuteration (Appendix A). Further optimization revealed that heating at 230 °C for 152 h in the presence of ammonia resulted in high deuteration: 95.4% at the backbone and 76.1% at the side chain, giving an overall average of 80.9% (Entry 6). Conversely, sodium hydroxide (Entry 3) and acetic acid (Entry 2) suppress side-chain deuteration. These findings therefore suggest that the observed effects of acidic/basic additives are not solely due to pH or solubility, and that ammonia may play a catalytic role.

In contrast to alanine (2), leucine (4) exhibits a decrease in side chain deuteration upon the addition of ammonia, resulting in a low overall deuteration level (13.2%) (Appendix A, entry 2). The overall deuteration rate increased to 35.3% when acetic acid was used (Entry 3). Similar trends were observed for both valine (3), with acetic acid addition increasing the deuteration from 13.3% to 77.8% for valine (Appendix A, entry 2).

Phenylalanine (6) and histidine (17), bearing aromatic and imidazole rings, respectively, exhibited high deuteration levels under the standard 200 °C/24 h condition (Appendix A). In contrast, tryptophan (7, Appendix A), which contains an indole ring, decomposed under the same conditions. When ammonia was added and the reaction was conducted at lower temperatures, the decomposition was suppressed. Optimal deuteration was achieved at 170 °C, yielding 93.3% backbone deuteration and 73.1% average incorporation (Appendix A, entry 2). Whereas tyrosine (10, Appendix A), which possesses a phenolic hydroxyl group, decomposed at 200 °C but showed no reactivity at 140 °C, indicating the need for further condition optimization. Therefore, we planned experiments at temperatures between 140 °C and 200 °C with the addition of acids or bases. Because tyrosine has low solubility, it remained largely as a solid powder even when mixed with D_2_O. To address this, acids or bases were added in amounts equimolar to twice the molar quantity of tyrosine. As a result, when acetic acid was added and the mixture was heated at 170 °C, mean deuteration level of 51.4% was achieved (Appendix A, entry 2).

Serine (8), with a hydroxyl-containing side chain, decomposed above 170 °C (Appendix A, entry 2) and showed only limited reactivity at 100 °C, with an average deuteration level of 7.7%, accompanied by partial decomposition (Entry 3). Threonine (9), another hydroxyl-containing amino acid, underwent oxidation or degradation even at 100 °C (Appendix A, entry 5). Because threonine is not normally decomposed at 100 °C, this can be considered as Pt/C accelerating the decomposition. To suppress this degradation, the catalyst was changed from the conventional Pt/C to Ru/C. Under these modified conditions (100 °C for 24 h), decomposition was successfully suppressed (Appendix A, entry 6). However, the deuteration level remains low (3.3%), indicating that further optimization of the reaction conditions is necessary.

The acidic amino acids, aspartic acid (11) and glutamic acid (12), as well as their corresponding amides, asparagine (13) and glutamine (14), were found to decompose at 200 °C (Appendix A). A previous report has suggested that Pd catalysts can effectively deuterate aliphatic compounds [33]. These findings indicate the necessity of exploring alternative metal catalysts.

Basic amino acids, such as lysine (15), which possess amino groups in the side chains, undergo decomposition even at 100 °C (Appendix A, entry 2). Amines are generally known to poison metal catalysts, although partial deuteration of arginine (16) has been observed (Entry 4). Future studies should therefore explore the use of acid additives to suppress decomposition and enhance deuteration efficiency.

Cysteine and methionine, which have sulfur in their side chains, could not be deuterated using our method; the cause of this phenomenon remains unknown (19 and 20, Appendix A). The NMR results indicated that most of the cysteine was present in an unreacted form, but small amounts of cystine and other byproducts were formed by the deuteration reaction. The same reaction was performed without the Pt/C catalyst, and the same by-products were formed when the catalyst was added. Therefore, it is reasonable to assume that the byproducts were generated by heating, and that the Pt/C catalyst did not contribute to them. On the other hand, methionine was also not deuterated and no byproducts were found to be formed (methionine is probably more thermally stable than cysteine).

In addition, the chemical compositions of the used Pt/C catalysts were analyzed using XRF method (the details of the XRF measurements are described in the Appendix A). Figure 3 shows the XRF spectra of the Pt/C catalyst before and after the deuteration experiments. The XRF spectrum of the Pt/C used for methionine deuteration had a large sulfur peak, whereas the XRF spectrum of the Pt/C used for cysteine deuteration had a small sulfur peak. These results indicated that methionine strongly coordinated to the Pt/C surface, whereas cysteine hardly coordinated to Pt. It is reasonable to suppose that, because the p*K*a value of the thiol group in cysteine is approximately 8.3, which is protonated in the reaction mixture, the coordination ability of cysteine to Pt is probably weak.

Although cysteine and methionine both contain sulfur atoms in their side chains, there are significant differences in their thermal stabilities and strengths of their interactions with platinum. In conclusion, the deuteration of cysteine and methionine could not be achieved, but on the other hand, this study provided important chemical insights into the deuteration reaction and sulfur-containing amino acids.

### 3.2. Chiral Separation of Deuterated Tryptophan

It is well known that small molecules possessing chiral centers may yield stereochemically distinct products when subjected to nucleophilic substitution reactions such as SN2, due to the inversion of the configuration at the reactive center [34]. It is reasonable to suppose that stereo inversion occurs during the substitution of hydrogen atoms directly bonded to the chiral α-carbon of the amino acid backbone with deuterium. In order to investigate whether deuteration alters the optical activity of the amino acids synthesized using the simplified deuteration apparatus, we measured the optical rotation of deuterated L-tryptophan.

As commercially available L-tryptophan is in its naturally occurring non-deuterated form, it is designated as h-L-Trp in this study to clearly distinguish it from deuterated tryptophan (d-Trp). Specifically, we have compared the specific rotation ([α]D^20^) of h-L-Trp with that of d-Trp, which was obtained by deuteration at 170 °C for 24 h (Table 2). The measured specific rotation of h-L-Trp was consistent with reported literature values (−30.5 to −32.5°), whereas d-Trp exhibited a markedly reduced optical rotation. This significant decrease suggests that the deuteration reaction induced partial epimerization at the chiral α-carbon, resulting in near-racemization of the sample.

Based on these findings, we further investigated the enantiomeric composition of deuterated tryptophan using chiral column chromatography to achieve enantiomeric separation.

As an initial step toward obtaining purified deuterated D-form of tryptophan (d-D-Trp), enantiomeric separation was performed using chiral column chromatography. Figure 4 shows the HPLC profiles of h-L-Trp (top) and tryptophan which we deuterated (bottom). The enantiomers were separated as free amino acids and analyzed using an online fluorescence detection system (350 nm). Because the peak at approximately 36 min was attributed to L-tryptophan, the peaks at approximately 31 and 36 min in the deuterated tryptophan data were attributed to D- and L-form tryptophan, respectively. These two peaks were clearly separated, and their respective intensities were nearly identical. Evaluation of the fluorescence peak areas of D- and L-form tryptophan suggested that D- and L-form were present at a ratio of 1:1. The ratio of D- and L-form was estimated to be 47.5:52.5, based on the obtained optical rotation spectra, and 46.6:53.4, using NMR spectroscopy. Therefore, the fluorescence peak area ratio on the HPLC system is not accurate enough to calculate optical isomer ratio of D- and L-form tryptophan.

### 3.3. Physicochemical Characterization of Deuterated Tryptophan

Tryptophan is an amino acid of particular interest in the fields of nutraceuticals and pharmaceuticals because it serves as a biosynthetic precursor for the neurotransmitters serotonin and melatonin [35,36]. However, tryptophan is known to exhibit relatively high photodegradability compared with other amino acids because of the breakdown of its indole ring upon UV absorption [37]. For example, formulations and supplements containing tryptophan may undergo degradation under light or oxidative conditions, potentially reducing their efficacy and generating harmful by-products [38]. Tryptophan degrades slowly in acidic conditions [39,40].

In this study, we have evaluated the effects of deuteration on the acid stability of tryptophan. The progress of degradation was monitored by measuring the changes in fluorescence intensity at 339 nm, the characteristic emission wavelength of tryptophan. Figure 5 and Appendix A show the time-dependent changes in the fluorescence spectrum of tryptophan in 6N HCl aqueous solution at 80 °C, and the inset shows the fluorescence intensity changes at 339 nm as a function of reaction time. Because it takes around 60 min after preparing the experimental solution at room temperature for the temperature to stabilize at 80 °C, the fluorescence spectrum data prior to 60 min were excluded from the reaction rate analysis. The fluorescence intensity of tryptophan decreased with increasing reaction time; however, there was no significant difference between the decomposition reactions of protonated and deuterated tryptophan. Although various kinds of decomposition products were generated by the decomposition reaction of tryptophane in 6N HCl solution [41,42], we assumed that the decrease in the fluorescence intensity at 339 nm correlated with tryptophan decomposition. Since chloride anions, which are present in large amounts in the reaction mixture, strongly contribute to the tryptophan decomposition reaction [41], the decomposition rate constant can be calculated according to the following equation (considering it as a pseudo-first-order reaction):lnC_t_ = −*k*_obs_*t* + lnC_0_(1)
where *k*_obs_ is the pseudo-first-order reaction rate constant (h^−1^) and C_t_ and C_0_ are the tryptophan concentrations at reaction times t = t and 0, respectively. The *k*_obs_ values were estimated to be 0.15 h^−1^ for h-Trp, 0.17 h^−1^ for d_40_-Trp (40% deuterated Trp), and 0.15 h^−1^ for d_70_-Trp (70% deuterated Trp), respectively. As mentioned above, since a wide variety of decomposition products must be produced in this reaction, obtained *k*_obs_ values may not be appropriate as reaction rate constants. However, it is clear that there was no significant difference in the *k*_obs_ values. It is reasonable to assume that the concentration of chloride anions in the reaction mixture strongly contributes to the tryptophan decomposition reaction, and the degree of deuteration of tryptophan has no discernible impact on the reaction. That is, isotope effects were negligible in the tryptophan decomposition reaction in the 6N HCl solution.

It is well known that tryptophan is degraded by ultraviolet light through a free radical pathway [37,43], and light shielding is necessary to inhibit its degradation. Deuteration makes tryptophan easier to handle if it is more resistant to UV degradation. In this study, we investigated the effects of isotopes on degradation resistance to ultraviolet light at a wavelength of 254 nm.

Figure 6 and Appendix A show the time-dependent fluorescence change of tryptophan under UV light irradiation in a 100 mM KCl solutions, and the inset shows the fluorescence intensity changes at 354 nm as a function of UV light irradiation time. Although the fluorescence intensities of tryptophan decreased with increasing UV light irradiation time, the fluorescence intensity changes stopped after approximately 20 min of irradiation (0.87 J/cm^2^), after which the fluorescence intensity changed again. Therefore, it is reasonable to assume that the fluorescence intensity of tryptophan changes bi-exponentially.

Previous reports have suggested that the UV degradation of tryptophan is mono-exponential [44,45], but our results are clearly different. Our findings may be attributed to the detailed evaluation of tryptophan UV degradation reactions at irradiation doses of 0–2.0 J/cm^2^. Assuming that the UV degradation of tryptophan is a pseudo-first-order reaction [44], the value of *k*_obs_ can be calculated using Equation (1).

The *k*_obs_ values obtained are listed in Table 3. As the degree of tryptophan deuteration increases, the *k*_obs_ value of the UV degradation reaction decreases. Note that the plots of the UV degradation reaction of d_40_-Trp do not appear to be bi-exponential; therefore, it is possible that the calculated *k*_obs_ values of d_40_-Trp are not as reliable as those of the others. In addition, the activation energy (*E*_a_) was estimated using the Arrhenius equation given in Equation (2).*k*_obs_ = *A* exp (−*E*_a_/*RT*)(2)
where *A*, *R*, and *T* are the pre-exponential factors of the Arrhenius equation, universal gas constant, and temperature, respectively. All UV degradation reactions were performed under the same conditions, with the only major difference being the degree of tryptophan deuteration. Assuming that this pre-exponential factor was the same under these conditions, the difference in *E*_a_ between protonated and deuterated tryptophan was calculated using the Arrhenius equation. According to the calculations, the activation energy difference between these UV degradation reactions was estimated to be approximately 0.48–0.74 kcal/mol. Based on density functional theory (DFT) calculations, the activation energy difference in the hydrogen/deuterium transfer reaction from anisole to methoxy radicals was estimated to be 1–1.5 kcal/mol [46]. The activation energy difference obtained in this experiment is similar to that of the hydrogen/deuterium transfer reaction. Because the dissociation energy of the van der Waals interactions is less than 1 kcal/mol, the stabilizing effect of tryptophan deuteration on the activation energy of the UV degradation reaction was comparable to the van der Waals interaction force. In summary, it can be concluded that deuteration renders tryptophan more resistant to UV light.

### 3.4. Structural Comparison Between Protonated and Deuterated Tryptophan Crystals

Deuterated tryptophan (d-Trp) and its chloride (d-TrpCl) crystallized in this study were racemic mixtures of the D- and L-forms (d-D/L-Trp and d-D/L-TrpCl), as confirmed by single-crystal X-ray diffraction. The crystal structure of d-Trp was dissolved in monoclinic P2_1_/c with one tryptophan molecule to form a racemate (Z = 4) (Figure 7, Appendix A). D/L-tryptophan adopts a zwitterionic form and forms a dimeric structure through a charge-assisted hydrogen bond (N5–H5a … O2) via the crystallographic inversion center. Two additional hydrogen bonds at the amino group (N5–H5c…O1 and N5–H5b…O2) link the tryptophan dimer along the b- and c-axes, respectively. Its crystal structure is similar to that of nondeuterated tryptophan (h-Trp) [47,48,49]. The differences in the bond lengths and angles of non-hydrogen atoms between d-Trp and h-Trp amount with maximum deviations of up to 0.01 Å and 0.86°, respectively (Appendix A). The measured values showed a small variation, but this difference was within the standard deviation (3σ range of the experimental error). Similarly, no significant differences were observed between the hydrogen bond lengths of d-Trp and h-Trp (Appendix A). Deuterated tryptophan chloride (d-TrpCl) crystallized from a racemic mixture as a racemic conglomerate. The crystal structure was determined to be monoclinic *P*2_1_ with one tryptophan cation and one chloride anion in the asymmetric unit. One chloride anion and three tryptophan cations are linked by hydrogen bonds (O1-D1…Cl16ta, N1-D5A…Cl16, N1-D5B…Cl16, N1-D5C…Cl16), resulting in three-dimensional hydrogen bond networks (Figure 8, Appendix A). The crystal structure was similar to that of non-deuterated tryptophan chloride crystals (h-TrpCl) [50]. There were no significant differences in the bond lengths, angles, and hydrogen bond lengths between d-TrpCl and h-TrpCl; the maximum deviations of 0.002 Å (bond lengths) and 0.26° (bond angles) (Appendix A). For the crystal structure refinement, the deuterium (or hydrogen atom) positions and thermal parameters were refined without any structural restraints. The differences in the C-D, N-D, and O-D bond lengths between d-TrpCl and h-TrpCl were within the standard deviation. In this study, hydrogen–deuterium exchange did not affect the bond lengths, angles, or intermolecular interactions in the crystal structures of tryptophan.

Some X-ray crystallographic studies have reported differences before and after deuteration. For example, in the case of potassium dihydrogen phosphate, the lattice constant gradually increases as the deuterium content increases [51]. In the case of protein, haloalkane dehalogenase from Xanthobacter autotrophicus (XaDHL), differences were observed in the structure of the active sites of deuterated-XaDHL [52]. Aspartic acid is displaced from its position in non-deuterated-XaDHL and rotates to form a hydrogen bond with histidine. These results suggest that the significant structural changes caused by deuteration mainly occur at hydrogen bonding sites. That is, X-ray crystallography cannot distinguish between deuterated and non-deuterated molecules whatsoever absent much more specific isotope effects of hydrogen bonds. 

## 4. Conclusions

In this study, we aimed to establish an efficient deuteration method for 20 proteinogenic amino acids, including their side chains, based on a platinum-on-carbon (Pt/C)-catalyzed hydrogen–deuterium exchange reaction. By optimizing the key reaction parameters, such as temperature, reaction time, additives, and metal catalysts, we significantly improved both the selectivity and stability of the process.

Nine amino acids achieved main-chain deuteration levels exceeding 90%, and six amino acids exhibited average deuteration levels above 70% when side chains were also included. Notably, valine, phenylalanine, and histidine—amino acids whose deuterated forms remain costly in the commercial market as of 2025—were efficiently and reproducibly synthesized using this method, highlighting its practical utility. In addition, decomposition-prone amino acids, such as serine and threonine, were successfully stabilized by careful temperature control and catalyst selection.

A particularly striking finding was the significant decrease in optical rotation observed for deuterated amino acids, indicating substantial racemization during the reaction. This underscores the need for enantiomeric separation to preserve the optical purity. Indeed, the chiral separation of tryptophan using a chiral column enabled the successful synthesis and isolation of deuterated D-tryptophan, a compound that is not commercially available.

Evaluation of the physicochemical properties revealed that while no clear isotope effect was observed for acid stability (tested in 6N HCl), a marked improvement in photostability was achieved. This suggests that deuteration is a promising strategy to enhance the photostability of amino acids.

Single-crystal X-ray diffraction analysis of deuterated tryptophan showed no significant changes in the bond lengths, bond angles, or intermolecular interactions upon hydrogen–deuterium substitution.

Collectively, these findings have established a practical and versatile platform for the stable supply of deuterated amino acids and peptides. This method has strong potential for application not only in neutron-based structural analysis and pharmacokinetic studies but also in the development of next-generation pharmaceuticals, functional foods, and biomaterials.

## Figures and Tables

**Figure 1 bioengineering-12-00981-f001:**
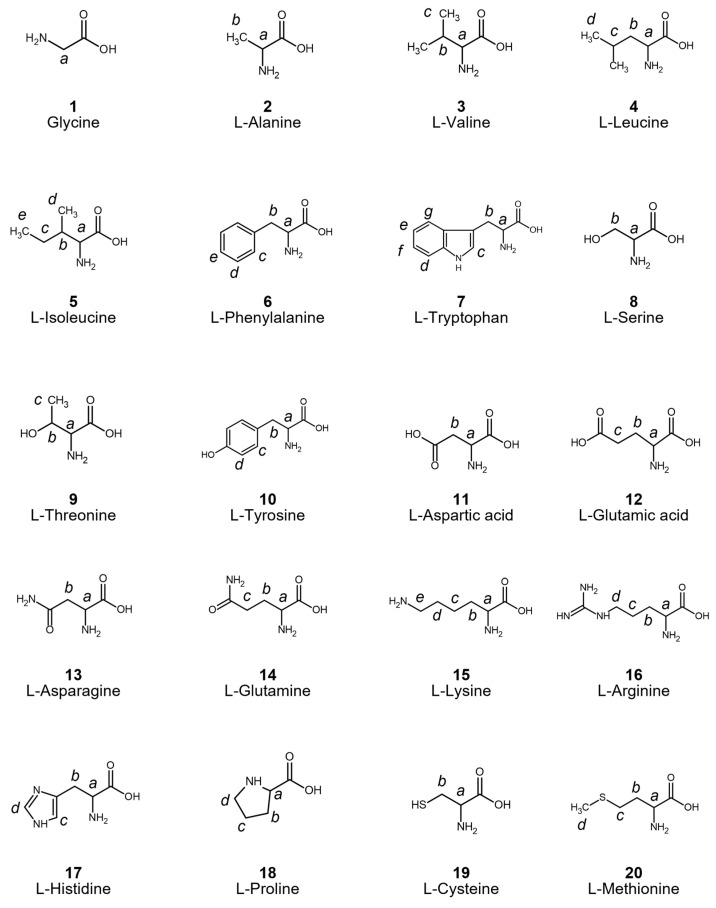
Samples of 20 standard amino acids.

**Figure 2 bioengineering-12-00981-f002:**
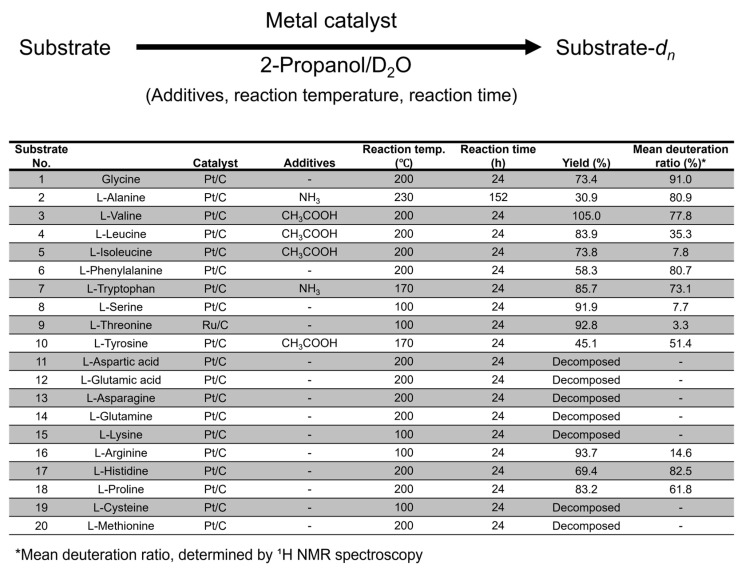
Deuteration results of 20 standard amino acids using acid/base additives under various temperatures. Each amino acid (1 g) was reacted in a mixture of 2-propanol (4 mL), catalyst, additive and D_2_O within the reaction vessel.

**Figure 3 bioengineering-12-00981-f003:**
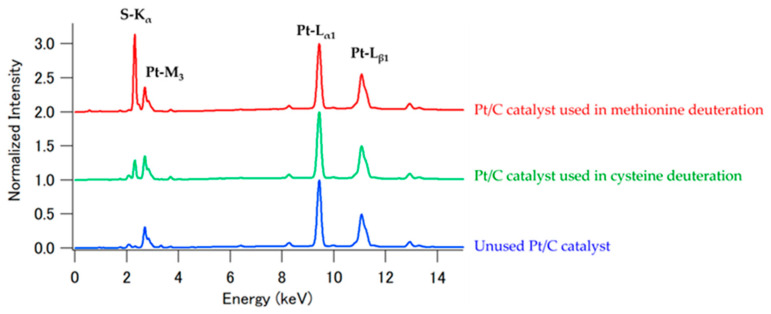
The result of the XRF analysis performed on Pt/C samples.

**Figure 4 bioengineering-12-00981-f004:**
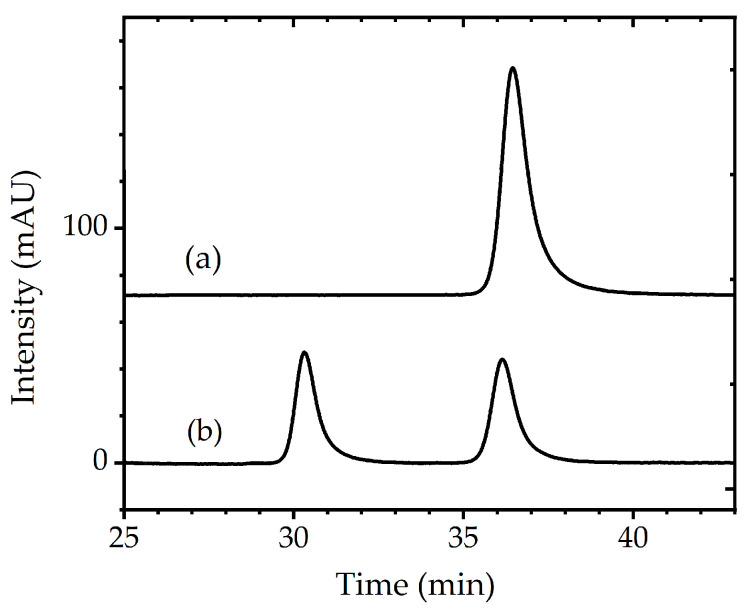
Enantiomeric separation using chiral column chromatography. (**a**) Commercial L-tryptophan (h-L-Trp); (**b**) Deuterated tryptophan. Detection was performed with excitation at 250 nm and emission at 350 nm.

**Figure 5 bioengineering-12-00981-f005:**
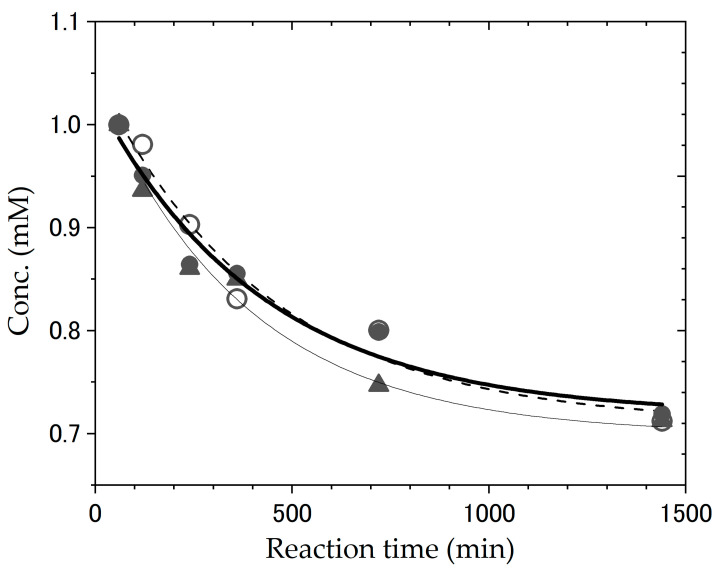
Time-dependent fluorescence intensity of tryptophan in acidic aqueous solution. Open circles represent commercially available hydrogenated tryptophan (h-Trp), closed triangles represent 40% deuterated tryptophan (d_40_-Trp), and closed circles represent 70% deuterated tryptophan (d_70_-Trp) synthesized in this study using a simple deuteration apparatus. Dashed, thin, and solid lines indicate the fitted curves for each dataset. The vertical axis indicates tryptophan concentration (mM), and the horizontal axis represents reaction time (min). Fluorescence measurements were performed with an excitation wavelength of 290 nm and an emission wavelength of 339 nm. The sample solutions were prepared at room temperature and required approximately 60 min to reach and stabilize at 80 °C; therefore, data collected prior to 60 min were excluded from the kinetic analysis.

**Figure 6 bioengineering-12-00981-f006:**
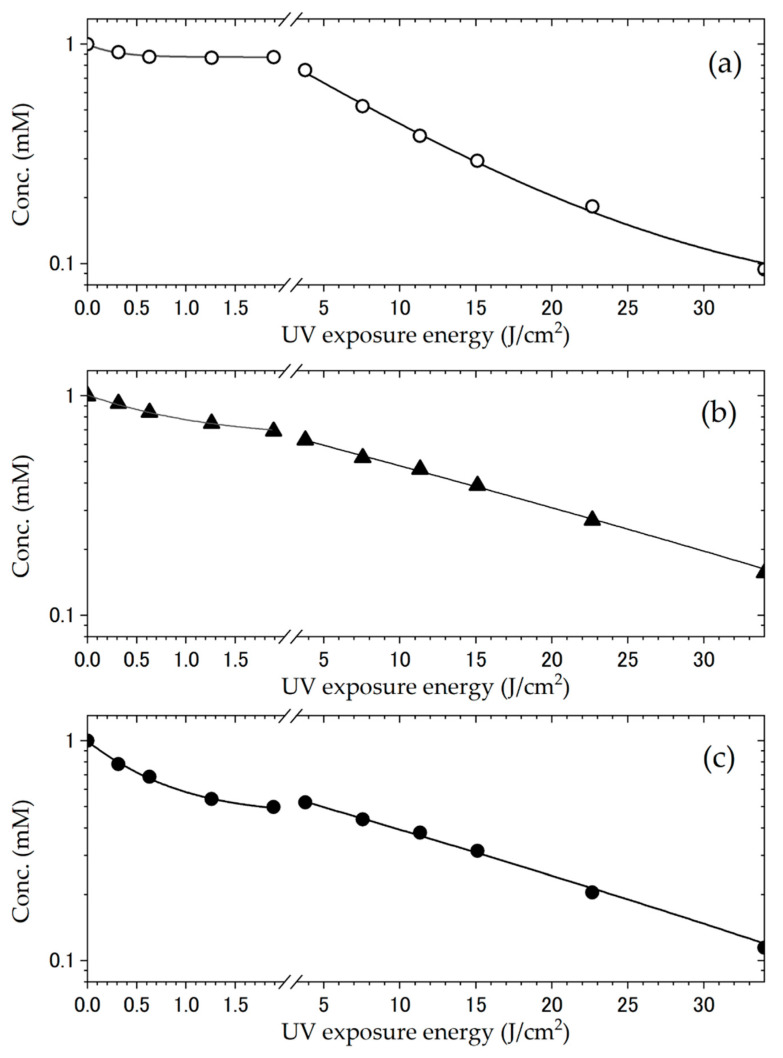
Photodegradation of tryptophan under UV irradiation. (**a**) Commercial tryptophan (h-L-Trp); (**b**,**c**) 40% and 70% deuterated tryptophan (d_40_-Trp and d_70_-Trp, respectively) synthesized in this study using a simple deuteration apparatus. The graphs show the time-dependent changes in the fluorescence intensity under UV light at 254 nm (62.94 mJ/min/cm^2^) in 100 mM KCl aqueous solution. Fluorescence was measured with an excitation wavelength of 290 nm and an emission range of 295–450 nm.

**Figure 7 bioengineering-12-00981-f007:**
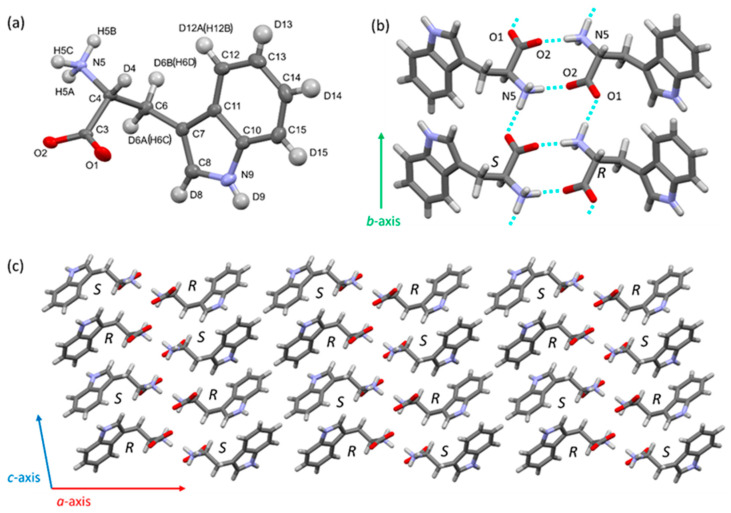
(**a**) Crystal structures of the deuterated tryptophan (d-Trp) as ellipsoid model with a probability of 50%. (**b**) Hydrogen bond (N5–H5a … O2) between (S)-tryptophan and (R)-tryptophan molecules forms a dimer structure. Another hydrogen bond (N5–H5c … O1) linked the tryptophan dimers along the b-axis. (**c**) Crystal packing of tryptophan.

**Figure 8 bioengineering-12-00981-f008:**
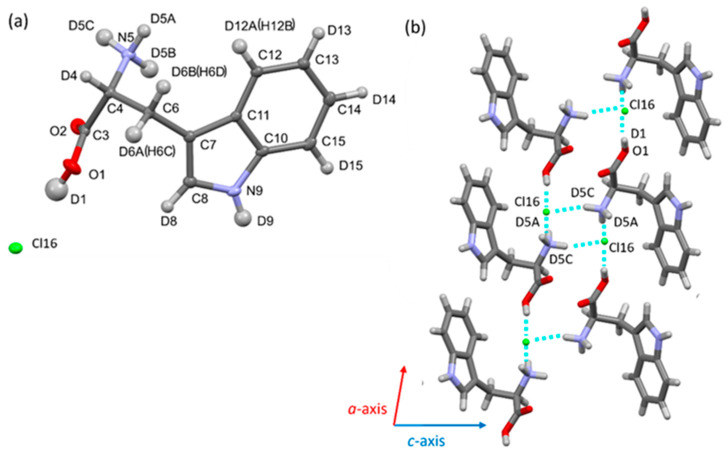
(**a**) Crystal structure of the deuterated tryptophan chloride (d-TrpCl) as ellipsoid model with a probability of 50%. The crystals were obtained as racemic conglomerate, the structure analyzed crystal consists of (R)-tryptophan. (**b**) Tryptophane cations and chloride anions are liked to form three-dimensional hydrogen bond network.

**Table 1 bioengineering-12-00981-t001:** Performance comparison of each deuteration method for amino acids.

Reaction Temperature	Racemization	Deuterated Site	Catalyst
37 °C	Non-racemizing	Partially deuterated (*α*- and/or *β*-position)	Enzyme [18,19,20]
70–135 °C	Non-racemizing	Partially deuterated (*α*-position)	Ru catalysts [14,15]
Room temperature	Racemizing	Partially deuterated (*α*-position)	Organocatalyst [23]
100–230 °C	Racemizing	Fully deuterated	Pt/C (This work)

**Table 2 bioengineering-12-00981-t002:** Polarimetric measurement results of tryptophan.

Sample	[α]D^20^ (deg·mL·g^−1^·dm^−1^) ± SD (20 °C, 598 nm)
L-Tryptophan (h-L-Trp)	−32.5925 ± 0.0199
Deuterated tryptophan (d-Trp)	−1.6525 ± 0.2656

**Table 3 bioengineering-12-00981-t003:** Summary of the obtained *k*_obs_ and activation energy difference values for the UV degradation reaction.

Sample	*k*_obs_ (First Reaction) and Δ*E*_a_	*k*_obs_ (Second Reaction) and Δ*E*_a_
h-L-Trp	0.230 min^−1^	0.0061 min^−1^
d_40_-Trp (40% deuterated)	0.065 min^−1^, 0.74 kcal/mol	0.0024 min^−1^, 0.61 kcal/mol
d_70_-Trp (70% deuterated)	0.080 min^−1^, 0.55 kcal/mol	0.0027 min^−1^, 0.48 kcal/mol

## Data Availability

The original contributions presented in this study are included in the article. Crystallographic data have been deposited at the CCDC under 2466294-2466296 and 2466306. Further inquiries can be directed to the corresponding authors.

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
