# Peer review of "Development of the Direct Deuteration Method for Amino Acids and Characterization of Deuterated Tryptophan"

_bioengineering, 2025, doi:10.3390/bioengineering12090981_

Round 1
Reviewer 1 Report
Comments and Suggestions for Authors
H/D exchange is a fundamental transformation that is challenging at non-acidic C-H bonds. Methods to accomplish this transformation from cheap deuterium sources and operate in high yield, or with good selectivity, are particularly useful for medicinal chemistry. Shibazaki et al present a method for H/D exchange of amino acids using Noble metal catalysts and D2O as a cheap deuterium source. Some further editing is required before the work is fit for publication. The results are, on the whole, a modest advance. But they do put some good experimental data on photostability of Trp into the literature and provide updated conditions for deuteration of each standard amino acid. Hopefully these writing changes are quick to accomplish after which the work is suitable for publication.
Major:
1 - The main claim of the work is that, “[we have] developed an efficient deuteration method that targets all 20 proteinogenic amino acids, including their side chains.” However, their method does not work with many amino acids. While use of additives was able to overcome come of the competing decomposition pathways, no methods were delivered for deuteration of Cys and Met. Given the subsequent focus on Trp, these limitations may not preclude publication. But they should still be discussed in a more accurate and responsible fashion.
2 - The author’s present the X-ray structure of d-D/L-Trp as if they are testing an unknown. This is not the case. As they themselves write, ‘The differences in the bond lengths and angles of non-hydrogen atoms between d-Trp and h-Trp …was within the standard deviation.’ That is, X-ray crystallography cannot distinguish between d- and h-forms of molecules whatsoever absent much more specific equilibrium isotope effects of H-bonds.
3 - Reactions use a 95/5 mixture of D2O-iPrOH, which implies 97.5% deuterium content. How then do the author interpret enrichment beyond a statistical outcome? Do protons preferentially partition onto the amines? Is there a reason the authors did not deploy deuterated iPrOD instead?
Minor
1 - Do the authors know what the current commercial routes to these per-deuterated amino acids are?
2 - Authors describe Pd-based approaches to deuterate amino acids. Biocatalytic methods have also been studied recently, notably from Naryan, Buller, and Hai. These methods include retention of stereochemistry and even work with Trp, which is a focus here.
https://pubs.acs.org/doi/full/10.1021/jacs.2c00608
https://chemistry-europe.onlinelibrary.wiley.com/doi/full/10.1002/cbic.202300561
3 - Author’s write, ‘Recently, neutron scattering has emerged as a powerful complementary method…’ While neutron scattering still advances, the technique itself is decades old.
4 - Author’s write, ‘Although cysteine and methionine have similar molecular structures.’ this is simply untrue. They share sulfur as a common element but otherwise have very different chemical properties.
Reviewer 2 Report
Comments and Suggestions for Authors
The manuscript presented for review is interesting and serious methodologically experimental work, which might be potentially useful for medicinal chemistry researchers. I recommend the manuscript for publication after addressing a number of recommendations and clarifications.
1) Unfortunately, I didn't find any copies of experimental NMR spectra. Given the importance of these data for determining the deuteration level (one of the main claimed results of the work), key copies of NMR spectra should be added to Supplementary materials.
2) It is not entirely clear from the results what the advantage of the new method is. A comparison table with literary methods should be added to reflect this fact and to more clearly demonstrate the novelty of the work.
3) Moreover, optimization tables and figures with its discussions should be transferred to the main article.
4) CCDC numbers for obtained crystal structures should be added to Figures 6 and 7 or its footnotes for clarity.
5) The abstract section should be reduced leaving only the stated results of the work.
6) I think that substrates should be numbered for more convenient discussion in the text of the work.
Author Response
Please see the attachment.
Referee comment _2
The manuscript presented for review is interesting and serious methodologically experimental work, which might be potentially useful for medicinal chemistry researchers. I recommend the manuscript for publication after addressing a number of recommendations and clarifications.
1) Unfortunately, I didn't find any copies of experimental NMR spectra. Given the importance of these data for determining the deuteration level (one of the main claimed results of the work), key copies of NMR spectra should be added to Supplementary materials.
We appreciate the reviewer's comment on this point. We added the NMR data for amino acids with the highest deuteration level to Supplementary materials.
2) It is not entirely clear from the results what the advantage of the new method is. A comparison table with literary methods should be added to reflect this fact and to more clearly demonstrate the novelty of the work.
We have developed an efficient deuteration method for 13 amino acids, including their side chains, on a gram scale. We further investigated the enantiomeric composition of deuterated tryptophan using chiral column chromatography to achieve enantiomeric separation. As a result, the chiral separation of tryptophan using a chiral column enabled the successful synthesis and isolation of deuterated D-tryptophan, a compound that is not commercially available. These are the benefits of our newly developed deuteration methods.
Several studies have addressed this topic: In the study of amino acids deuteration using enzymes, an enzyme catalyzed rapid deuteration of alpha and beta hydrogen atoms in amino acids without racemization. In the study of amino acids deuteration using Ru catalysts, Ru catalysts catalyzed effective deuteration of alpha hydrogen atoms in amino acids without racemization. In the study of amino acids deuteration using organocatalysts, organocatalysts catalyzed effective exchanges of alpha hydrogen atoms in amino acids with racemization. Ru catalysts catalyzed effective deuteration of alpha hydrogen atoms in amino acids without racemization.
We agree that this point requires clarification, we summarized these experimental findings in Table 1 and have added the following text to the Introduction section: Ruthenium catalysts appear to successfully catalyze the deuteration of alpha hydrogen atoms in amino acids while preventing racemization. Organocatalysts accelerated the effective deuteration of alpha hydrogen atoms in amino acids at room temperature through racemization [23]. In the case of enzymic reaction, alpha and beta hydrogen atoms in amino acids were rapidly deuterinated by an enzyme without racemization [20].
3) Moreover, optimization tables and figures with its discussions should be transferred to the main article.
We have added the optimization tables (Table 1), along with their discussions, to the main article.
4) CCDC numbers for obtained crystal structures should be added to Figures 6 and 7 or its footnotes for clarity.
We wish to thank the Reviewer for this comment. Crystallographic data have been deposited at the CCDC under 2466294-2466296 and 2466306. Therefore, we have added a sentence “Crystallographic data have been deposited at the CCDC under 2466294-2466296 and 2466306.” into the Data Availability Statement section.
5) The abstract section should be reduced leaving only the stated results of the work.
We thank the reviewer for this helpful suggestion. We have shortened the first part of the summary and revised it to: “Proteins and peptides are vital biomolecules, and deuterated amino acids are increasingly applied in areas such as drug discovery, metabolic tracing, and neutron scattering studies.”
6) I think that substrates should be numbered for more convenient discussion in the text of the work.
We thank the Reviewer for this helpful suggestion. We have numbered the substrates and added the corresponding numbers to all tables, figures, and the main text.

Reviewer 3 Report
Comments and Suggestions for Authors
The present manuscript describes the synthetic protocol towards the preparation of deuterated aminoacids using platinum on carbon (Pt/C) as a catalyst. The deuteration composition and levels have been studied by the NMR spectroscopy, while the HPLC technique was used to monitor the enantioselectivity of the processes. Composition of the recovered heterogeneous catalyst was studied by the XRF spectroscopy. The resistance of the selected deuterated aminoacids towards UV irradiation was studied as well. Remarkably, the single crystal X-ray diffraction data have been collected for several deuterated hydrochloride products and compared to the known normal (non-deuterated) analogues. Deuteration techniques have numerous applications in (bio)chemistry, where generation of selectively C-H deuterated products is of great importance. Therefore, this manuscript can be recommended for publication after correction of minor issues listed below.
- Figure 1 appears confusing. The reaction in the top shows “Pt/C”, while there were other metals tested (e.g. Ru/C). Where is the “asterisk” mentioned in the footnote? The text in parentheses is poorly readable. Please consider reorganizing this table in a “classic” form as the consecutive list of entries.
- Was the blank test (in the absence of a catalyst) performed? If yes, please include it in Figure 1 as well.
- The Schemes in Schemes S3-S7 are broken with overlapped text/figures. Consider inserting them as bitmap images, but not vector ones.
- It is known that for biomedical applications the residual amount of a transition metal catalyst in the product should not exceed certain ppm levels (typically very low ones). Was the leaching of platinum and its persistence in the deuterated products studied?
- The experimental protocol 2.3 (line 122) states “200-70°C”. Should it be 200-270? Please check.
Round 2
Reviewer 2 Report
Comments and Suggestions for Authors
Authors have completed all comments as per suggestion. Therefore, Manuscript will be accepted in present form.